# Peer review of "Health Disparities Investigator Development through a Team-Science Pilot Projects Program"

_ijerph, 2023, doi:10.3390/ijerph20075336_

Round 1

Reviewer 1 Report

The approach to include representatives and investigators from the minority ethnicities to reduce health disparities is certainly an area that needs lot more attention. This is a great start. 

Although there is good data the manuscript requires revision in the methods, results and conclusion. Please see below some of the comments and suggestions below: 

line 38 - reads like a conclusion. Can be revised. 

line 46 - revised to state as a fact would be appropriate in the introduction

line 61 - Stating the limitations of the traditional approach could be included in the introduction. 

The methods section should discuss the team-science model and what it involves. Not the results or observations of the approach. 

Are the sub sections part of the team-science approach? If so would be helpful to include a sentence describing the components of the approach in the first paragraph of the methods section

Why is this paragraph included here instead of section 2.3?

Why is this a separate sub-section? This paragraph continues to explain the unique aspects of the mentoring program which is the purpose of sub-section 2.3

line 153 - group the percentages appropriately. Could be shown as a Venn-diagram. Early stage investigators as the overlap and number of ethnic minority and women.

Numbers do not add up. Could be represented as a chart. 

line 164 - conclusion statement. 

The methods and results section should be revised to state the appropriate details. The challenges due to COVID-19 pandemic should be included in the discussion and not the results but the data point should remain under the results. The data in table 1 can be represented as a pie diagram or Venn-diagram. In table 3 the columns could be abbreviated if possible to be neater. Or if the columns and rows could be switched so the numbers could read better. 

Author Response

The approach to include representatives and investigators from the minority ethnicities to reduce health disparities is certainly an area that needs lot more attention. This is a great start.  Although there is good data the manuscript requires revision in the methods, results and conclusion. Please see below some of the comments and suggestions below: 

line 38 - reads like a conclusion. Can be revised.

            We greatly appreciate the reviewer’s helpful suggestions and comments. We agree with the reviewer and have revised the introduction section. This manuscript is a conference proceeding report outlining a unique approach to Pilot Projects Program and early-stage investigator development. As such, this does not represent an original research article, but we used the format of an original research article to organize the presentation.

line 46 - revised to state as a fact would be appropriate in the introduction.

We have revised the introduction section to provide more detail as to the health disparities experienced by NHOPI and Filipinos.

Revised Introduction

Health disparities in Hawai‘i are substantial, especially among Native Hawaiians (NH), other Pacific Islanders (OPI) and Filipinos (1-3). Compared to other ethnic groups in Hawai‘i, NHOPI and Filipinos suffer disproportionately higher rates of obesity (4; 5), hypertension (6; 7), diabetes (8-10), chronic kidney disease (11), cardiovascular and cerebrovascular diseases (12; 13), and breast and lung cancers (14-16). They also have the lowest life expectancy of any other racial and ethnic groups in Hawai‘i (3; 17).  Research efforts to reduce the disparities are limited by a shortage of skilled and well-trained investigators, especially from these ethnic groups (18-21). The paucity of NH, OPI and Filipinos participating in research as investigators (22) follows a shortage of minority faculty members in the biomedical and health-related fields (23) and the limited health disparities research mentoring of these groups. Further, the traditional model of a single-investigator led research project is increasingly insufficient for addressing the complex social, economic, and environmental challenges that contribute to health disparities. Professional development organized via mentoring networks, individual development plans (IDPs), research education, and training in community engagement combined with access to pilot project funding has been shown to increase success in health disparities research (24-26). At the 2022 Research Centers in Minority Institutions (RCMI) Consortium National Conference, we shared lessons learned in health disparities investigator development by reporting on the successes and challenges of our RCMI Pilot Projects Program (PPP) from 2017–2022 in regard to health disparities research within the State of Hawai‘i. This conference proceedings paper presents our approach and related outcomes.

line 61 - Stating the limitations of the traditional approach could be included in the introduction.

We agree with the reviewer and have moved the sentence stating the limitations of the traditional approach to single investigator led research projects into the introduction section.

The methods section should discuss the team-science model and what it involves. Not the results or observations of the approach.

            We agree that explaining our team-science model is important and have added to the method section the following:

A team-science approach was encouraged via the following methods: (1) marketing and weighting pilot project selection based upon multidisciplinary and multi-professional investigator team participation; (2) encouraging demonstration of community relevance and community participation in pilot project development and selection; (3) providing assistance in multidisciplinary and multi-professional investigator team development during Research Infrastructure Core consultations and with other RCMI sites; (4) demonstrating examples of successful team science during Mentoring Bootcamp sessions; and (5) encouraging team-science participation in institutional health science academic promotions. In addition to encouraging team science, other innovative methods were used and described below. We also describe outcome metrics, such as resultant scientific presentations, peer-reviewed publications, extramural funding, and examples of successful pilot projects that resulted in behavioral changes in community participants in the Results section.

Are the sub sections part of the team-science approach? If so would be helpful to include a sentence describing the components of the approach in the first paragraph of the methods section.

We believe the methods subsections are important to describe as complementary and supportive of our team-science approach. We have included a sentence describing these supportive methods and have removed redundant sentences.

Why is this paragraph included here instead of section 2.3? 

We have removed paragraph prior to section 2.3 as it was redundant with the mentorship plan.

Why is this a separate sub-section?

This paragraph continues to explain the unique aspects of the mentoring program which is the purpose of sub-section 2.3 

“Section 2.4. Expanding the Mentorship and Reviewers Network” highlighted the need for mentorship within and outside of our institution which we distinguished from “Section 2.3 Personalizing the Mentoring Bootcamp” which focuses on training opportunities. These are two separate features of the PPP.

line 153 - group the percentages appropriately. Could be shown as a Venn-diagram. Early stage investigators as the overlap and number of ethnic minority and women. 

The number of minority and female investigators overlap significantly.  Additionally, there were overlaps within the different colleges/schools/departments where these investigators held appointments. The purpose of this table is to show the ethnic diversity of the investigators, inclusion of female investigators, and ability to attract early-stage investigators in units across the university. Although the data could be shown as a Venn-diagram of minority and female investigators, this would overly restrict the features noted above. We have slightly modified the table to reduce confusion regarding the study population.

Numbers do not add up. Could be represented as a chart.

As noted above, there is considerable overlap with multiple Center/College/School investigators on each project. The intent is to emphasize multiple measures of diversity in the participating early-state investigators.

line 164 - conclusion statement.

We thank the reviewer for this comment and have revised the sentence to say, “Extramural funding and publication data are shown in Table 2, along with the return on investment (ROI) in terms of additional extramural funding generated from pilot grants.”

The methods and results section should be revised to state the appropriate details. The challenges due to COVID-19 pandemic should be included in the discussion and not the results but the data point should remain under the results.

We agree with the reviewer and have moved the details of this section to the discussion.

The data in table 1 can be represented as a pie diagram or Venn-diagram.

As mentioned above, counts of minority and female investigators and represented colleges/schools/departments for these investigators overlapped. The purpose of this table was to show the general diversity of the investigators, inclusion of female investigators, and ability to attract early-stage investigators across the campus. Although the data could be shown as a Venn-diagram or pie diagram, the gestalt of the investigator diversity would be lost. Therefore, we have kept the general form of the table.

In table 3 the columns could be abbreviated if possible to be neater. Or if the columns and rows could be switched so the numbers could read better.  

We agree that Table 3 does not display the data as clearly as we would have liked. We have decided to remove Table 3 and instead provide the abridged results in the text.

Reviewer 2 Report

I congratulate the authors for the topic of study and its development. The inclusion of minorities is relevant and necessary. However, I miss the incorporation of the gender perspective in the study and in the proposals. In this regard, I suggest that you consult the SAGER Guide, which will provide you with clear and practical information that will facilitate your work.

Author Response

I congratulate the authors for the topic of study and its development. The inclusion of minorities is relevant and necessary. However, I miss the incorporation of the gender perspective in the study and in the proposals. In this regard, I suggest that you consult the SAGER Guide, which will provide you with clear and practical information that will facilitate your work. 

We greatly appreciate the reviewer’s helpful suggestions and comments. We agree with the reviewer that the inclusion of minorities and women are essential to diversity of the scientific workforce. We have read the SAGER Guide for reporting research subject enrollment. This manuscript is a conference proceeding report summarizing our approach to Pilot Projects Program (PPP) and early-stage investigator development. As this is a retrospective conference proceeding report which refers to mentored PPP investigators rather than enrolled study subjects, application of the SAGER Guide methodology does not appear warranted.

Reviewer 3 Report

The paper does not present scientific research nor the particular effects of the projects described. It is not a research paper but a rather vague description of the project with some data about project performance. As any detailed effects of health disparities research nor tightening the gap between minorities and the general population are described, I suggest including at least 2 case studies with real-world examples proving meeting project objectives.
The introduction is very brief, even though broad literature exists on the topic.

Author Response

The paper does not present scientific research nor the particular effects of the projects described. It is not a research paper but a rather vague description of the project with some data about project performance. As any detailed effects of health disparities research nor tightening the gap between minorities and the general population are described, I suggest including at least 2 case studies with real-world examples proving meeting project objectives.

We greatly appreciate the reviewer’s helpful suggestions and comments. As this manuscript was prepared for a special issue of the IJERPH to highlight the 2022 RCMI Consortium National Conference, our manuscript provides a detailed description of the PPP as a mechanism to foster health disparities investigator development and enhancement of health disparities research programs. We provide outcomes information summarizing multidisciplinary and multi-professional team-science with our communities of greatest health needs.  As suggested by the reviewer, we have provided two examples of Pilot Projects and their effects on community participants

MALAMA: Backyard Aquaponics to Promote Healthy Eating and Reduce Cardiometabolic Risk (feasibility study): Chung-Do (Office of Public Health Studies) and Ho-Lastimosa (College of Tropical Agriculture & Human Resources) received pilot funds to systematically test the acceptability and feasibility of backyard aquaponics workshops as a health intervention (27; 28). Using a single group pre-post design, 10 NH families (n=21 individual participants) from Waimanalo participated in a health promotion intervention, named MALAMA (Mini Ahupuaa for Lifestyle And Meaai [food] through Aquaponics). In addition to being an acronym, the word “malama” in Hawaiian means, to “take care,” “preserve”, “protect” and “nurture.” The PPP awardees developed and implemented six family-based workshops over three months. Individuals from the community who had prior experience in aquaponics from the previous years served as Lima Kokua, or peer leaders, by sharing their lessons learned with the participants and providing support throughout the workshops. There was consistent attendance through the six workshops. Although this was a feasibility study, favorable changes in health behaviors and outcomes were found. Fruit consumption significantly increased from 2.1 to 2.9 servings/day, and favorable changes were reported in vegetable and fish consumption.  There were also financial savings achieved through growing their own fresh fruits, vegetables and fish. A trend in reduction of systolic and diastoic blood pressure was also noted. Participants found workshops to be culturally acceptable, identified the relationship building aspects of the intervention as essential, and recommended the intervention be extended from three to six months.

Community Driven Approach to Mitigate COVID 19 Disparities in Hawai‘i's Vulnerable Populations: May Okihiro (Department of Pediatrics), Ruben Juarez (Department of Economics) and Alika Maunakea (Department of Anatomy, Biochemistry & Physiology) received pilot funds to assess COVID-19 vaccine hesitancy among vulnerable adults living in Hawai‘i (29). The investigators assembled a multidisciplinary team of academic and community investigators, along with long-standing community partners across Hawai‘i, to form a collaborative called the Pacific Alliance Against COVID-19 (PAAC) to participate in the National Institutes of Health Rapid Acceleration of Diagnostics-Underserved Populations (RADx-UP) Initiative (30). Partners included the Accountable Healthcare Alliance of Rural Oahu (AHARO), a consortium of five federally qualified community health centers (FQHCs), and K-12 public schools (kindergarten through grade 12) that serve communities on three islands with large proportions of NH and OPI. The study found that for 1,124 adults residing in a region with one of the lowest COVID-19 vaccination rates in Hawai‘i revealed that race/ethnicity was not directly associated with the probability of vaccine uptake. Instead, a higher degree of trust in official sources of COVID-19 information increased the probability of vaccination by 20.7%, whereas a higher trust in unofficial sources decreased the probability of vaccination by 12.5% per unit of trust (31). These results revealed a dual and opposing role of trust on COVID-19 vaccine uptake. Interestingly, NH and OPI were the only racial/ethnic group to exhibit a significant positive association between trust in and consumption of unofficial sources of COVID-19 information, which explained the vaccine hesitancy observed in this indigenous population. These results offer novel insight relevant to COVID-19 mitigation efforts in ethnic minority populations.

The introduction is very brief, even though broad literature exists on the topic.

We agree with the reviewer and have revised the introduction section to include details of the health disparities experienced by NHOPI and Filipinos. Although the literature on the importance of community engaged research and the importance of mentorship for under-represented minority investigators is extensive, the publication size limits led us to focus on background information most relevant to the NHOPI and Filipino populations.

Reviewer 4 Report

Dear Authors,

The manuscript entitled “Health Disparities Investigator Development Through a Team-Science Pilot Projects Programdeals with an interesting and important topic; however, it is not really a scientific paper but a final project report. That is the major problem with the manuscript.

Besides, other issues also have risen, among others:

- the abstract does not contain the methods of the research and the results;

- the first two sentences of the Introduction is part of the solution, so it should not be at the very beginning of the manuscript;

- the manuscript discusses health disparities but almost nothing is presented about the nature and prevalence of those disparities;

- the discussion of relevant literature is missing;

- the Materials and Methods section is not really about methods of the research but it introduces the program itself under investigation; and

- the results presented are limited to the given project (number of publications, grants submitted and awarded, funding, etc.) but the real results of these kinds of programs lie in the behavioral change of citizens to lessen health disparities; however, this topic is not even mentioned in the manuscript.

Author Response

The manuscript entitled “Health Disparities Investigator Development Through a Team-Science Pilot Projects Program” deals with an interesting and important topic; however, it is not really a scientific paper but a final project report. That is the major problem with the manuscript.

We greatly appreciate the reviewer’s helpful suggestions and comments. As this manuscript was prepared for a special issue of the IJERPH to highlight the 2022 RCMI Consortium National Conference, our manuscript provides a detailed description of the PPP as a mechanism to foster health disparities investigator development and enhancement of health disparities research programs. We provide outcomes information summarizing multidisciplinary and multi-professional team-science with our communities of greatest health needs. Although we used a research article format for organization, we have tried to make it clear that this is a descriptive conference proceeding report rather than a research article.

Besides, other issues also have risen, among others: 

- the abstract does not contain the methods of the research and the results; 

The abstract has been updated accordingly as follows (within the 100 word limit according to the Special Edition manuscript submission instructions:

Profound health disparities are widespread among Native Hawaiians, other Pacific Islanders, and Filipinos in Hawai‘i. Efforts to reduce and eliminate health disparities are limited by a shortage of investigators trained in addressing the genetic, socioeconomic, and environmental factors that contribute to disparities. In this conference proceeding report from the 2022 RCMI Consortium National Conference, we describe our mentoring program with an emphasis on community-engaged research. Elements include our encouragement of a team-science, customized Pilot Projects Program (PPP), a Mentoring Bootcamp, and a mentoring support network development. We describe the successes and challenges of the PPP awardees past and present along with potential solutions.

- the first two sentences of the Introduction is part of the solution, so it should not be at the very beginning of the manuscript;

We have reworded the introduction section to remove the two sentences that we believe to be solutions and conclusions. We have also provided relevant background on health disparities specific to NHOPI and Filipinos.

Revised Introduction

Health disparities in Hawai’i are substantial, especially among Native Hawaiians (NH), other Pacific Islanders (OPI) and Filipinos (1-3). Compared to other ethnic groups in Hawai‘i, NHOPI and Filipinos suffer disproportionately higher rates of obesity (4; 5), hypertension (6; 7), diabetes (8-10), chronic kidney disease (11), cardiovascular and cerebrovascular diseases (12; 13), and breast and lung cancers (14-16). They also have the lowest life expectancy of any other racial and ethnic groups in Hawai‘i (3; 17).  Research efforts to reduce the disparities are limited by a shortage of skilled and well-trained investigators, especially from these ethnic groups (18-21). The paucity of NH, OPI and Filipinos participating in research as investigators (22) follows a shortage of minority faculty members in the biomedical and health-related fields (23) and the limited health disparities research mentoring of these groups. Further, the traditional model of a single-investigator led research project is increasingly insufficient for addressing the complex social, economic, and environmental challenges that contribute to health disparities. Professional development organized via mentoring networks, individual development plans (IDPs), research education, and training in community engagement combined with access to pilot project funding has been shown to increase success in health disparities research (24-26). At the 2022 Research Centers in Minority Institutions (RCMI) Consortium National Conference, we shared lessons learned in health disparities investigator development by reporting on the successes and challenges of our RCMI Pilot Projects Program (PPP) during 2017–2022 in regard to health disparities research within the State of Hawai‘i. This conference proceedings paper presents our approach and related outcomes.

-  discussion of relevant literature is missing;  

We have now provided relevant background in the introduction section on health disparities specific to NHOPI and Filipinos.

- the Materials and Methods section is not really about methods of the research but it introduces the program itself under investigation;  

This manuscript is a descriptive conference proceeding report outlining a unique approach to Pilot Projects Program and early-stage investigator development. We have revised the methods section to better capture this descriptive nature. 

- the results presented are limited to the given project (number of publications, grants submitted and awarded, funding, etc.) but the real results of these kinds of programs lie in the behavioral change of citizens to lessen health disparities; however, this topic is not even mentioned in the manuscript. 

 The goal of this manuscript was to highlight the Pilot Projects Program process and how it can assist in early-stage investigator development. As suggested by the reviewer on how the PPP can impart behavioral change in citizens, we provide two examples of successful Pilot Projects that resulted in behavioral changes among its community participants and participating health agencies.

Round 2

Reviewer 3 Report

My suggestions and comments from the first review have been sufficiently addressed. The version presented meets the minimum publication requirements.

Author Response

We thank the reviewer for their helpful comments and suggestions. This reviewer had no further comments.

"My suggestions and comments from the first review have been sufficiently addressed. The version presented meets the minimum publication requirements."

Reviewer 4 Report

Dear Authors,

The manuscript entitled “Health Disparities Investigator Development Through a Team-Science Pilot Projects Program” has been improved compared to its earlier version; however, there are a few issues to be addressed:

- the abstract still does not contain the results;

- it is not clear what “marketing and weighting pilot project selection” means;

- the manuscript discusses health disparities and now some pieces of information are presented about the nature and prevalence of those disparities in the Introduction section; however, more specific data are still missing as well as data on the number of investigators;

- two examples are mentioned in the manuscript related to the behavioral outcome (thank you), but an overall impact study would provide a real scientific value of this paper; and

- there are some typos in the text (see, e.g., lines 30 and 37).

Author Response

We thank the reviewer for the additional insights. Our responses/revisions are shown in italics below.

Changes in the manuscript are highlighted in green.  Prior changes to the manuscript after the first round of reviews are lighted in yellow.

- the abstract still does not contain the results;

We had originally kept the results section brief in regards to prior instructions to authors (100 word limit was mentioned in the Special Issue manuscript submission instructions). We have slightly expanded the results to incorporate distinct results and conclusions. Please see below for the additional information in the revised abstract.

“…In 2017–2022, we received 102 PPP preproposals. Of these, 45 (48%) were invited to submit full proposals, and 22 (19%) were awarded (8 basic biomedical, 7 clinical, 7 behavioral). Eighty-three percent of awards were made to early-career faculty (31% ethnic minority, 72% women). These 22 awards generated 77 related publications; 84 new grants were submitted of which 31 were awarded with a resultant return on investment of 5.9. From 5 to 11 investigators were supported by PPP awards each year. Robust usage of core services was observed. Our descriptive report (as part of a scientific conference session on RCMI specialized centers) focuses on a mentoring vehicle and shows how it can support early-stage investigators pursuing  careers in health disparities research.”

- it is not clear what “marketing and weighting pilot project selection” means; 

Please see lines 78-82 of the edited revision (and below) where we clarify that this comment is made in response to publicizing our team science approach to potential PPP investigators when soliciting and reviewing applications. The text now reads:

“(1) utilizing a multidisciplinary and multi-professional investigator team to review and evaluate pilot projects;..”

- the manuscript discusses health disparities and now some pieces of information are presented about the nature and prevalence of those disparities in the Introduction section; however, more specific data are still missing as well as data on the number of investigators;

While we appreciate the reviewer’s interest in overall health disparities impact, our descriptive report (as part of a scientific conference session on RCMI specialized centers) focuses on the structure and function of a pilot project program (PPP) as an early-stage investigator mentoring vehicle.  The report is intended to provide helpful guidance which may be used at other institutions seeking to develop such a mechanism for investigator mentoring. We have provided examples, including projects (as requested by the reviewer) addressing community behavioral and/or policy matters with some related outcomes. To provide an overall community impact report, especially when the focus is on a PPP to develop early-stage investigators is beyond the scope of the conference presentation and submitted paper.

Nonetheless, we have added information on the number of investigators participating as awardees in the PPP each year and provided the following note after describing the PPP examples requested by the reviewer: “Behavioral and other health changes in the wider community and population are difficult to measure, given these were pilot projects.”

- two examples are mentioned in the manuscript related to the behavioral outcome (thank you), but an overall impact study would provide a real scientific value of this paper; and

We appreciate the question and have incorporated our response to this question in the information provided above. We have added the following text:

 “Behavioral and other health changes in the wider community and population are difficult to measure, given these were pilot projects.”

- there are some typos in the text (see, e.g., lines 30 and 37).

We appreciate having our attention drawn to the typos and believe we have corrected those in the edited text.